# CRISPR/Cas Technology in Insect Insecticide Resistance

**DOI:** 10.3390/insects16040345

**Published:** 2025-03-26

**Authors:** Qiuchen Xu, Mingyun Wang, Jiahui Zeng, Hangzhen Sun, Xiaoqi Wei, Hui Jiang, Xuping Shentu, Dan Sun

**Affiliations:** Key Laboratory of Microbiological Metrology, Measurement & Bio-Product Quality Security, State Administration for Market Regulation, College of Life Science, China Jiliang University, Hangzhou 310018, China; xxxuqqqiu@163.com (Q.X.); wmycjlu@163.com (M.W.); zjh17857686679@163.com (J.Z.); 18505585736@163.com (H.S.); weixiaoqi12368@163.com (X.W.); 18953543931@163.com (H.J.)

**Keywords:** insects, insecticide resistance, CRISPR/Cas, target site

## Abstract

Chemical and biological insecticides are crucial in modern agricultural, forestry, and public health pest management, and are involved in protecting crops, preserving human health, and maintaining ecological balance. However, the misuse of insecticides has led to increased insect resistance, mainly due to altered expression or mutations in insect target genes that encode proteins involved in pesticide recognition, binding, and detoxification. The emergence of CRISPR/Cas gene-editing technology has deepened our understanding of the mechanisms by which insects adapt to and resist insecticides. This article summarizes the progress of this technology in the prevention and control of agricultural pests and public health pests in different arthropods.

## 1. Introduction

Insects are an important part of biodiversity. They provide important ecosystem services such as pollination and pest control but also cause damage as disease vectors and plant pests [1,2,3]. Scientists have attempted to tackle these key issues through biological and chemical methods. However, extensive and widespread use of insecticides has led to pest resistance as a consequence, which impedes pest control effectiveness [4,5,6]. As the global population continues to rise, there will be a huge demand for food in the near future [7]. Crop pests are considered a serious challenge to agriculture production and threaten food security [8]. According to incomplete statistics, global annual crop losses caused by insect pest damage are up to hundreds of billions of dollars [9]. Public health pests often serve as vectors or carriers that transmit various bacteria and viruses posing threats to human health [10,11,12]. Biological and chemical insecticides serve as the two primary pest management strategies in agriculture, safeguarding crop yield and quality. However, the unscientific and non-standard application of these pesticides also causes increasing pest resistance [13,14,15]. The evolution of insect resistance to insecticides is frequently related to the expression alterations or mutations in the target site [16,17,18,19]. Certain key genes in insects exhibit altered expression levels upon exposure to pesticides, resulting in significant changes in the concentration of corresponding intracellular protein products [20]. These proteins often serve as primary targets for insecticides, such as Acetylcholinesterase (AChE) that serves as the primary target for commercial pesticides, particularly organophosphate and carbamate insecticides [21]. In many insects that develop resistance to pesticides, such as the DDT-resistant malaria vector *Anopheles gambiae*, the activity of Glutathione S-transferases (GSTs) increases significantly. This enhanced activity plays a crucial role in the resistance mechanism, enabling these insects to more effectively eliminate or neutralize pesticide molecules that enter their bodies [22]. For example, the high expression of *CYP6F1* in *Culex pipiens pallens* can confer resistance to deltametharin [23]. Gene mutations can alter the amino acid sequence at the interaction sites between proteins and insecticides, resulting in target modifications that typically diminish insecticide efficacy and produce exceptionally high levels of resistance [24,25].

The rapid advancement of gene-editing technologies has facilitated studies that explore and manipulate specific gene functions in insects [26]. The first prevalent technology for targeted genome editing used zinc finger nucleases (ZFNs) [27]. ZFNs consist of zinc finger proteins that specifically recognize DNA sequences fused with Fok I nucleases. This setup induces double-strand breaks (DSBs) in DNA, activating the cellular DNA damage repair mechanisms and triggering gene mutations (Figure 1A) [28,29]. As early as 2002, ZFNs were applied to insect species. For instance, Bibikova M and colleagues mutated the color gene in Drosophila using ZFNs [30]. Beumer K and others produced mutated offspring of fruit flies through the same method [31]. Subsequently, ZFNs were also applied in other insect models, such as *Caenorhabditis elegans* [32] and *Danaus plexippus* [28]. Although ZFNs played a pivotal role in early insect gene-editing research, their relatively complex preparation process and high costs have largely limited their widespread application. Subsequently, transcription-activator-like effector nucleases (TALENs) were developed, which achieved the specific recognition and cleavage of DNA sequences with the help of Fok I nucleases (Figure 1B). TALEN exhibits higher specificity and lower toxicity than ZFNs when cutting the same sequence [33,34]. Scientists innovatively applied an enhanced TALEN technology to achieve rapid and efficient modifications of the Drosophila genome. They utilized the “modular assembly” strategy to combine smaller, recognizable DNA sequence units into larger, more complex protein structures capable of recognizing and cleaving specific DNA sequences, thereby constructing specific TALENs targeted at the yellow gene and a novel autosomal gene. Subsequent mRNA injections into Drosophila embryos yielded results indicating that 31.2% of the fertile F0 generation carried heritable modifications of the yellow gene, completing the entire process in just one month [35].

The CRISPR/Cas genetic modification platform represents a groundbreaking advancement in gene editing, fostering novel opportunities for the creation of biopesticides and the investigation of pesticide resistance mechanisms [36]. The CRISPR/Cas system is currently in widespread use, including various types such as CRISPR/Cas12a (also known as Cpf1), CRISPR/Cas13a, and CRISPR/Cas13b, each with unique advantages and application potentials. Among the many members of the CRISPR/Cas system, CRISPR/Cas9 stands out as the most prominent tool. Its efficiency, precision, and ease of use have quickly made it the mainstream tool in the field of gene editing. Notably, the key difference between CRISPR/Cas9 and ZFN and TALEN lies in its use of short RNA sequences as specificity recognition units, activating the Cas9 enzyme to induce double-strand breaks in target DNA [37] (Figure 1C). Whilst, compared with ZFNs and TALENs, CRISPR/Cas has demonstrated significant advantages in the study of insecticide resistance. Its high efficiency, simplicity, low cost, and ease of use enable researchers to quickly and accurately cut and edit target genes, thereby precisely manipulating genes related to insecticide resistance in a short period of time, observing the effects of genetic changes on the phenotype of organisms, and revealing their roles and mechanisms in drug resistance [38,39,40,41,42]. Moreover, the changes at the genomic level produced by CRISPR/Cas are stable and heritable, allowing mutated genes to transfer to the next generation [43,44,45].

The CRISPR/Cas technology has garnered significant attention in the field of entomology, largely due to the successful genomic editing of the classic model organism *Drosophila melanogaster* [46]. Subsequently, CRISPR/Cas has been effectively utilized in various insects, including *Plutella xylostella* [47], *An. gambiae* [48,49], and *Bombyx mori* [50]. The rapid advancements in omics and molecular biology in recent years have led to an increasing application of CRISPR/Cas in entomological studies. However, there is currently a lack of a comprehensive review article that systematically explores and integrates the application of CRISPR/Cas technology in insect insecticide resistance. This article reviews the application and prospects of the CRISPR/Cas system concerning insecticide resistance in Diptera, Lepidoptera, and Hemiptera. The emergence of CRISPR/Cas technology presents new strategies and methodologies for exploring insect gene functions and field pest control. As technology continues to advance and improve, CRISPR/Cas-based pest management strategies will play a more crucial role in agricultural production, contributing significantly to food security, ecological safety, and human health.

## 2. Applications of CRISPR/Cas9 in Insecticide Resistance

The studies that have used CRISPR/Cas9 in insecticide resistance have been generalized according to species from different insect orders and the types of gene editing (Figure 2). Detailed information has also been organized to facilitate a comparison (Table 1).

### 2.1. Diptera

#### 2.1.1. *Drosophila melanogaster*

As a classic model organism in biological research, *D. melanogaster* has consistently been at the forefront of genetic analysis, significantly contributing to the advancement of entomology [114]. The CRISPR/Cas9 technology serves as a crucial tool in understanding various genotypes, receptors, and ion channels that influence resistance formation in Drosophila. CYP genes are pivotal members of insect metabolic detoxifying enzymes, playing a vital role in the development of insect resistance. In *D. melanogaster*, *CYP6G1* and *CYP6G2* are key candidate genes involved in metabolism and resistance to multiple insecticides. Denecke utilized CRISPR/Cas9 to knockout *CYP6G1* and *CYP6G2* genes, revealing that the high expression of these two genes confers imidacloprid resistance [51]. Fusetto further demonstrated that *CYP6G1* is associated with the formation of insecticide oxidative metabolites [52]. Insect cell membrane ion channels are pivotal targets for a wide array of insecticides, with functional diversity being intrinsically associated with pest resistance. The kdr mutation in voltage-gated sodium channels (vgscs) is notably linked to resistance to prevalent insecticides in insects. The kdr mutation was successfully introduced into *D. melanogaster* using CRISPR/Cas9 and subsequently evaluated the sensitivity variations in these mutants to permethrin DDT and deltamethrin. Furthermore, they employed CRISPR-based allele drives to substitute the resistant kdr mutations with their susceptible wild-type alleles. This foundational validation paves the way for numerous potential applications, such as the targeted reversion of insecticide-resistant populations to a state of susceptibility [53]. A combination of genetic transformation and CRISPR/Cas9 genome-editing technology demonstrated that the transgenic *D. melanogaster* strains expressing P450 enzymes along with engineered mutations in the voltage-gated sodium channel (para) are associated with pyrethroid resistance [54]. The nicotine acetylcholine receptor (nAChR), a neurotransmitter-gated ion channel protein, is widely distributed in the central nervous system of insects and serves as a critical target for various insecticides. The G275E mutation in the nAChR *Dα6* subunit imparts resistance to spinosad, achieved through CRISPR/Cas9-mediated gene knock-in in *D. melanogaster* [55]. A systematic study of the nAChR gene family in *D. melanogaster* was conducted using CRISPR/Cas9 to explore the response mechanisms to three different classes of insecticides: neonicotinoids, spinosyns, and sulfoximines [56]. Recently, researchers developed a sensitivity gene drive at the acetylcholine esterase (Ace) locus in *D. melanogaster* using CRISPR/Cas9 technology, and they successfully replaced resistant mutation types with sensitive ones, potentially reversing pest resistance and restoring sensitivity to pesticides [57]. Similarly, CASPP ((5-chloro-1′-[(E)-3-(4-chlorophenyl)allyl] spiro[indoline-3,4′-piperidine]-1-yl}-(2-chloro-4-pyridyl)methanone) compounds represent a novel class of neuroactive insecticides that specifically target the Vesicular Acetylcholine Transporter (VAChT). Vernon integrated transgenic *D. melanogaster* lines with the GAL4/UAS system to enhance the transcription level of VAChT. In unison, they introduced the *VAChT^Y49N^* mutation using CRISPR/Cas technology and found that *D. melanogaster* develop resistance to CASPP when the expression of wild-type VAChT is upregulated or there is a specific point mutation (*VAChT^Y49N^*) [58]. ABC transporters play a pivotal role in insecticide resistance, facilitating the transport of various neuroactive insecticides. The knockdown of three genes, *Mdr65*, *Mdr49*, and *Mdr50*, of the ABC transporter in *D. melanogaster* using CRISPR/Cas9 revealed that the mutation of *Mdr65* increased the sensitivity to a range of neuroactive insecticides, whereas the responses of *Mdr49* and *Mdr50* varied depending on the specific insecticide. The combination of the CRISPR/Cas9 method and ABC inhibitor verapamil further confirms the role of ABC transporters in enhancing insecticide tolerance in *D. melanogaster* and unveils their tissue and substrate specificity [59].

Insects with GABA receptors, particularly those composed of the RDL (resistant to dieldrin) subunit, are critical targets for commonly used synthetic insecticides. Zhou examined the sensitivities of seven distinct *D. melanogaster* RDL point mutants to four meta-diamide and isoxazoline insecticides [60]. CRISPR/Cas9 technology has significantly contributed to understanding insecticide resistance in *D. melanogaster* by modifying key genes such as CYPs, nAChR, VAChT, ABC transporters, and ion channels, providing insights into resistance mechanisms and facilitating the development of new pest management strategies.

#### 2.1.2. *Aedes aegypti*, *Culex quinquefasciatus*, and *Anopheles gambiae*

Mosquitoes pose severe threats to human health by transmitting numerous diseases such as Zika, malaria, dengue, chikungunya, and filariasis [115,116,117]. These vector-borne viruses have emerged as a critical public health concern, affecting approximately 4 billion people worldwide. Vector control, including the use of insecticide-treated nets (ITNs) and indoor residual spraying (IRS), has proven effective in reducing malaria transmission in epidemic and hyperendemic areas. The application of insecticides is a crucial component of this initiative. However, the rise in insecticide resistance among mosquitoes and pathogens has led to the resurgence of these diseases [118].

Membrane proteins play a pivotal role in protecting cells from foreign substances, including drugs and toxins. Researchers have studied these proteins, particularly P-glycoprotein encoded by the *MDR49* gene, and found that they confer resistance to certain insecticides, such as ivermectin, and can influence the reproductive capabilities of mosquitoes like *Ae*. *aegypti* [119]. Other membrane proteins, such as cadherin (Cad) and membrane-bound alkaline phosphatase (mALP), are potential receptors for Cry toxins from Bacillus thuringiensis, although the exact relationship between these proteins and Cry toxin activities remains unclear.

Using the CRISPR/Cas9 system, scientists have successfully generated knockout mutants of these membrane proteins in mosquitoes. For instance, Pacheco generated mALP- and Cad-knockout mutants in *Ae. aegypti* [61], while Lan identified ten candidate nicotinic acetylcholine receptor (nAChR) subunits and created a mutant strain lacking the *Aaeα6* gene, which exhibited increased resistance to spinosad [120]. In addition to *Ae. aegypti*, *C*. *quinquefasciatus* is another global vector for various human and animal diseases [121,122]. Feng developed a specific Cas9/sgRNA expression toolkit for *C. quinquefasciatus*, providing significant support for the development of control technologies for this species [62]. Researchers have also used CRISPR genome-editing techniques to introduce mutations, such as the kdr mutation *L1014F* in *An. gambiae*, to investigate their effects on insecticide resistance [63]. Williams et al. confirmed that the *V402L* confers resistance to permethrin and DDT in *An. Gambiae* via the CRISPR/Cas9 system [64].

To prevent the transmission of mosquito-borne viruses, two primary strategies are proposed, i.e., reducing the capacity of vectors to harbor pathogens and suppressing insect populations through sterilization [123]. Membrane proteins and their mutations, studied using CRISPR/Cas9, play a crucial role in insecticide resistance and could be targets for developing new control measures against mosquito-borne diseases.

### 2.2. Lepidoptera

In recent years, CRISPR/Cas9 technology has introduced new tools for analyzing functional genes across various insect species. Among the Lepidoptera, *B. mori* is notable as the first agricultural insect to successfully undergo efficient genome editing using CRISPR/Cas9 for functional gene studies [26]. This order also encompasses several significant agricultural pests, such as *P. xylostella*, *H. armigera*, *S. exigua*, and *S. frugiperda*, which pose serious threats to crop yields. The emergence of resistance has rendered previously effective chemical control measures unreliable, complicating pest management considerably. By employing CRISPR/Cas technology to modify the specific genes within these pests, researchers can influence the growth, development, reproduction, and environmental adaptability of target insects, thus achieving pest control objectives.

#### 2.2.1. *Helicoverpa armigera*, *Helicoverpa zea*, *and Pectinophora gossypiella*

*H. armigera* is a notorious global pest that infests cotton crops. Researchers have identified cadherin (CAD) as the receptor for the Bt Cry1A toxin in *H. armigera*. Wang employed CRISPR/Cas9 technology to successfully knockout the *HaCad* gene in a susceptible *H. armigera* population, establishing a homozygous SCD-Cad knockout strain that exhibited 549-fold resistance to the Cry1Ac toxin than that of the susceptible stain [85]. Subsequently, they also knocked out two Rdl homolog genes (*HaRdl-1* and *HaRdl-2*), which encode critical subunits of insect γ-aminobutyric acid (GABA) receptors, the primary targets of cyclodiene and phenylpyrazole insecticides. The findings indicate that *HaRdl-1* and *HaRdl-2* play essential roles in determining the susceptibility of cotton bollworms to cyclodiene and other insecticides, exhibiting distinct pharmacological characteristics [86]. Furthermore, Wang discovered that the high resistance of *H. armigera* to the Bt toxin Cry2Ab is closely linked to a loss-of-function mutation in its ATP-binding cassette gene (*ABCA2*) [87]. In response, they developed two *HaABCA2* knockout strains, designated as SCD-A2KO1 and SCD-A2KO2. The results reveal that both knockout strains display extreme resistance to Cry2Aa and Cry2Ab toxins while showing minimal or no resistance to Cry1Ac toxin. Further investigations demonstrate that the resistance of these knockout strains to Cry2Ab is recessive, with the resistance locus located on the *HaABCA2* gene [88]. Zhang generated a series of homozygous knockout strains, including two single-gene knockout strains, in which the ATP-binding cassette (*ABCC2*) or *CAD* genes were deleted, and exhibited 512- or 396-fold Cry1Ac resistance ratio, respectively. Additionally, a double-gene knockout strain with *HaABCC2* and *HaCAD* was simultaneously deleted and showed 6273-fold resistance to Cry1Ac toxin, significantly greater than single-gene knockouts and confirming the interaction of *ABCC2* and *Cad* in *H. armigera* [89]. Cytochrome P450 plays a critical role in insecticide resistance in *H. armigera*. Wang used CRISPR/Cas9 genome-editing system to knockout a cluster of nine P450 genes, creating a homozygous mutant strain devoid of the *CYP6AE* cluster. This mutant strain exhibited reduced tolerance to esfenvalerate and indoxacarb. Furthermore, they confirmed that no single *CYP6AE* gene is essential for the survival of *H. armigera*. This finding significantly contributes to the future identification of candidate genes associated with insecticide resistance [124]. The Glutathione S-transferase (GST) gene family plays a significant role in insect detoxification mechanisms. Researchers found that the GST gene cluster in the North China strain of *H. armigera* underwent strong selection, indicating a significant resistance to *lambda*-cyhalothrin resistance. They knocked out a GST cluster using the CRISPR/Cas9 system, significantly increasing the susceptibility of *H. armigera* to *lambda*-cyhalothrin [90]. There are also cotton pests. Fabrick knocked out the *ABCA2* gene in *H. zea* and *P. gossypiella*, respectively, resulting in mutated populations resistant to Bt toxin Cry2Ab [91,92]. Research on *H. armigera* and related cotton pests using CRISPR/Cas9 technology has identified key genes involved in insecticide resistance, providing insights into the mechanisms of resistance and guiding strategies for pest management.

#### 2.2.2. *Plutella xylostella*, *Trichoplusia ni*, and *Tuta absoluta*

*T*. *ni* and *P*. *xylostella* are significant agricultural pests that cause serious damage to crops. Bt (Bacillus thuringiensis) can produce a series of insecticidal proteins for eliminating pests [125]. Guo provided the first in vivo reverse genetic evidence that *ABCC2*, *ABCC3*, *APN1*, *APN3a* and *ABCC1* serve as midgut functional receptors for Bt Cry1 toxins in *P. xylostella* [65,66,67]. Subsequently, Sun investigated the contributions of two ABC transport proteins (*ABCC2* and *ABCC3)* and two aminopeptidases N *(APN1* and *APN3a*) towards Bt Cry1Ac toxin resistance. Their findings indicated that double-gene knockouts significantly boosted resistance, especially a quadruple-gene knockout, which resulted in exceptionally high resistance, suggesting functional redundancy with mutual complementation among these receptors [68]. Additionally, functional redundancy between ABC transporters (*ABCC2* and *ABCC3*) and *PxmALP* also has been demonstrated by using CRISPR/Cas9-mediated multiple-gene knockout strategy [69]. The CRISPR/Cas9-induced knockout of non-receptor paralogs demonstrated that the increased expression of these genes is responsible for diminishing the fitness costs of Cry1Ac resistance [67]. Liu demonstrated that a mutation in either *PxABCC2* or *PxABCC3* alone is insufficient for high-level Cry1Ac resistance, necessitating mutations in both genes [70]. Zhao showed that dual-gene knockout of *PxABCC2* and *PxABCC3* significantly enhances resistance to Cry1Fa compared to single-gene knockout [71]. The CRISPR/Cas9-based mutation of *PxABCB1* in *P. xylostella* exhibited a 63-fold resistance to Cry1Ac toxin compared to the parental susceptible strain [72]. Ye and his team validated the potential role of the methionine aminopeptidase gene *PxMetAP1* in a cosmopolitan pest for Bt Cry1Ac toxin tolerance using CRISPR/Cas9 technology, which revealed a significant increase in susceptibility to Cry1Ac toxin [73]. Similar genes also exist in *Trichoplusia ni*, and Yang generated a homozygous strain of *T. ni* with frameshift mutations in *ABCA1* and *ABCA2* through CRISPR/Cas9 mutagenesis. Their findings indicate that the *ABCA2* mutants were highly resistant to Cry2Ab, whereas mutations in *ABCA1* do not exhibit the same level of resistance [97]. Ma also utilized CRISPR/Cas9 technology to validate the role of the resistance locus *ABCC2* in conferring resistance to the Bt toxin Cry1Ac in *Trichoplusia ni*. Subsequent hybridization experiments revealed that the F1-generation Cry1Ac-resistant strains exhibited higher Cry1Ac resistance levels compared to the *ABCC2*-knockout CRISPR strains, suggesting the presence of additional mutations that may be associated with Cry1Ac resistance [98]. The CRISPR/Cas9-mediated knockout of the *PxJHBP* gene exhibited increased sensitivity to Cry1Ac toxin, reduced lifespan, and decreased fertility. This highlights the crucial role of *PxJHBP* in Cry1Ac resistance and female reproductive regulation, offering new insights into designing insecticides or interference strategies targeting insect reproductive systems [74]. Researchers used a high-efficiency CRISPR/Cas9 system to knockout SE2 (short interspersed nuclear element (SINE—named SE2)) in the promoter of *MAP4K4* gene from the resistant *P. xylostella* strain, thereby altering the transcript levels of midgut genes linked with Cry1Ac resistance that showed an expression landscape similar to that of the susceptible *P. xylostella* strain [75]. The CRISPR/Cas9-mediated knockout of a midgut microRNA miR-8545 significantly increased the susceptibility of *P. xylostella* to Cry1Ac toxin [76]. The relationship between a mutation (*I1056M/F*) in the chitin synthase 1 (CHS1) gene of *P. xylostella* and benzoylureas (BPU) was verified in the *D. melanogaster* model using the CRISPR/Cas9 system [77]. Guo created a site-specific mutation of the m^6^A site in JH esterase (*PxJHE*) by the CRISPR/Cas9-mediated homology-directed repair (HDR) pathway in *P. xylostella*, resulting in a slight increase in the susceptibility to Cry1Ac protein [78].

There are many different genes that enhance the resistance of *P. xylostella* to different types of insecticides. Additionally, a reverse genetic approach was employed to confirm the association between the nAChR *α6* subunit and spinosad resistance in *P. xylostella* [79]. Sun identified a novel valine-to-isoleucine mutation (*V263I*) in the glutamate-gated chloride channel (GluCl) in the field strain of *P. xylostella*, demonstrating varied resistance levels to abamectin. They successfully established a *P. xylostella* knock-in strain expressing the homozygous *V263I* mutation via CRISPR/Cas9, which exhibited a high resistance (106.3-fold) to abamectin but significantly reduced fecundity [80]. Similarly, CRISPR/Cas9-mediated *D472N* substitution in the Rdl1 GABAR of *P. xylostella* confers low resistance to abamectin and endosulfan, with no effect on other GABAR-targeting insecticides [81]. Meta-diamide insecticides combat Lepidopteran pests by activating insect ryanodine receptors (RyRs). Wang first demonstrated the functional role of the *I4790M* mutation in *PxRyR* concerning meta-diamide resistance [82]. Jiang introduced the *I4790K* mutation in *PxRyR*, transforming sensitive populations into resistant ones, indicating a strong correlation of this mutation with resistance to various insecticides, including diamides [83]. *F1845Y* and *V1848I* are two mutations in the IVS6 fragment of the *P. xylostella* and *Tuta absoluta* sodium channel domains. To confirm their roles in indoxacarb resistance, Samantsidis employed CRISPR/Cas9 technology to create homozygous mutants carrying *F1845Y* or *V1848I* in *D. melanogaster*. Results indicated that both mutations conferred moderate resistance to indoxacarb, while the *F1845Y* homozygous mutant exhibited significant resistance to metaflumizone [108]. Certain insecticides, such as chlorantraniliprole and other diamide-class insecticides, disrupt insect muscle contractions by activating or inhibiting ryanodine receptors (RyRs), ultimately leading to insect mortality. Consequently, RyRs emerge as primary targets for these insecticides. These mutations were replicated in *D. melanogaster* by CRISPR/Cas9 and conducted toxicity assessments. Their findings reveal that the *G4946V* mutation significantly enhances resistance to flubendiamide and chlorantraniliprole of *T. absoluta*, while the *I4790M* mutation predominantly affects resistance to flubendiamide. These findings not only substantiate the significance of *RyR* mutations in diamide resistance but also shed light on the intricate ways in which these mutations can alter the binding affinity of different diamide insecticides, providing a deeper understanding of the molecular basis of resistance in pests [109]. The critical role of the *RyR* gene in conferring resistance to diamide insecticides was emphasized by further studies, underscoring the intricate interplay of multiple mutations in the evolution of this resistance [84].

Pests like *T. ni* and *P. xylostella* damage crops significantly. Bt toxins from Bacillus thuringiensis are effective against these pests. CRISPR/Cas9 studies have revealed that multiple genes, such as *ABCC2*, *ABCC3*, *APN1*, *APN3a*, and others, play roles in Bt toxin resistance through functional redundancy. Mutations in these genes enhance resistance to toxins like Cry1Ac and Cry1Fa. Similar genes and mutations are also found in *Trichoplusia ni*. CRISPR/Cas9 has been used to validate the roles of these genes and mutations in resistance, and to study their interactions. Additionally, genes like *PxJHBP*, microRNAs, and chitin synthase are also involved in resistance and reproductive regulation.

#### 2.2.3. *Spodoptera frugiperda*, *Spodoptera exigua*, and *Spodoptera litura*

*S. frugiperda*, *S. exigua*, and *S. litura* all belong to the family Noctuidae. *S. frugiperda* is one of the most damaging pests globally and has developed resistance to Cry proteins. The Bt vegetative insecticidal protein Vip3Aa serves as an effective substitute for Cry proteins in controlling several major pests that have not yet demonstrated practical resistance [126]. CRISPR/Cas is crucial for understanding how insecticides affect the physiological functions of pests. Based on the understanding of these toxins and the mode of action of insecticides, we can develop more sensitive biomarkers or monitoring tools for the early detection of drug resistance in pest populations and develop more precise management strategies [127]. Jin discovered that knocking out the transcription factor gene *SfMyb* reduced the susceptibility of *S. frugiperda* to Vip3Aa, and these results may facilitate advancements in the proactive monitoring and management of pest resistance to Vip3Aa [128]. Liu found that the knockout of the midgut-specific chitin synthase gene *SfCHS2*, mediated by a retrotransposon, resulted in a high level of resistance to Vip3Aa. This finding is vital for the monitoring and management of pest resistance to Vip3Aa in *S. frugiperda* [94]. Shi established a knockout strain of the nAChR *α6* subunit in *S. frugiperda* using a CRISPR/Cas9 system with dual sgRNAs to induce a substantial deletion of a DNA fragment. Their results demonstrated that the disruption of *Sfα6* leads to a remarked increase in resistance to spinosyns [95]. Xu discovered that *CYP304F1* mutated by CRISPR/Cas9 technology significantly diminished the resistance to β-cypermethrin and chlorpyrifos in *S. litura* [93]. These research findings provide new perspectives and methods for basic research on pest resistance. Researchers can further explore the molecular mechanisms, genetic laws, and the impact of environmental factors on drug resistance, providing a more solid theoretical basis for subsequent monitoring and management work.

*S. exigua* is a globally distributed pest. Zuo successfully introduced the RyR *G4946E* mutation into *S. exigua* using CRISPR/Cas9 technology. The mutant exhibited resistance to chlorantraniliprole, cyantraniliprole, and flubendiamide at levels of 223-fold, 336-fold, and over 1000-fold, respectively [99]. The CRISPR/Cas9-mediated knockout of P-glycoprotein (P-gp) significantly increased the susceptibility of *S. exigua* to abamectin and emamectin benzoate (EB), but not to spinosad, chlorfenapyr, beta-cypermethrin, carbosulfan indoxacarb, chlorpyrifos, phoxim, diafenthiuron, chlorfluazuron, chlorantraniliprole, or two Bt toxins (Cry1Ca and Cry1Fa) [100]. Zuo established a homozygous population (Seα6-KO), demonstrating that this gene is linked to resistance against spinosad and spinetoram [101]. Furthermore, Zuo introduced the *G275E* mutation of nAChR α6 found in palm thrips and whiteflies into *S. exigua* to validate the role of the *G275E* mutation in nAChR α6 in spinosyn resistance [102]. Wang identified nine nAChR subunits in *S. exigua*, namely, nAChR *α1*-*α7* and nAChR *β1*-*β2*, and created a deletion mutant (Seα1-KO) using CRISPR/Cas9 for the *Seα1* gene. Compared to the wild type, *Seα1-KO* exhibited significantly enhanced resistance to trifluoropyrimidine, dimehypo, and dinotefuran, while sensitivity to other insecticides remained unchanged [103]. Shi discovered significant yet differing enrichment of the *CYP9A* gene cluster between *S. exigua* and *S. frugiperda* through comparative genomics, which included species-specific duplications and sequence variations. CRISPR/Cas9 knockout experiments demonstrated that *CYP9A* contributes to the detoxification of plant defense compounds and various chemical insecticides. Additionally, the *CYP9A* gene cluster was closely associated with high pyrethroid resistance in field populations [96].

#### 2.2.4. *Ostrinia furnacalis* and *Chilo suppressalis*

*O. furnacalis* is a significant pest affecting crops such as corn throughout China, from Heilongjiang to Hainan. This species has developed a high level of resistance to the insecticidal crystals of Bt, particularly the Cry proteins Cry1A and Cry1F. This resistance is associated with mutations in the ATP-binding cassette (ABC) transporter subfamily C gene, ABCC2. Wang employed the CRISPR/Cas9 technique to successfully create a homozygous strain of *O. furnacalis* featuring an 8 bp deletion mutation in the *ABCC2* gene (OfC2-KO). Functional studies demonstrated a causal relationship between the truncated mutation of *OfABCC2* and increased resistance to the Cry1Fa toxin [104]. Furthermore, Jin confirmed that *OfCad* mediates Cry1Ac toxicity through both in vitro and in vivo experiments. Compared to the wild type, the *OfCad* knockout strain exhibited moderate resistance to Cry1Ac while maintaining sensitivity to other Cry proteins [129]. Additional studies targeting the ABC transporter subfamily G gene, *ABCG4*, revealed that mutations in this gene resulted in increased resistance to Cry1 toxins and affected larval development and population parameters [105]. The resistance of the *C. suppressalis* to tetraniliprole is increasing, and RyR double mutations (*I4758M* and *Y4667C*) provide greater resistance to chlorfenapyr than single mutations [106]. Further evidence has shown that, in a single RyR mutation, *Y4667D* contributes the most to resistance to tetraphenylproline, followed by *G4915E*. *Y4667C* and *I4758M* are roughly equal, while *Y4891F* contributes the least [107]. This knowledge will support the development of innovative pest management strategies to address the escalating issue of resistance.

### 2.3. Hemiptera

Hemipteran species possess specialized piercing–sucking mouthparts that enable them to ingest liquid food. In herbivorous hemipterans, these intricate mouthparts facilitate the efficient extraction of nutrients from the xylem or phloem of plants. This adaptive evolution has led to certain hemipterans becoming significant pests in global agriculture, resulting in substantial annual crop losses. The long-term dependence on chemical pesticides in agricultural settings has prompted many hemipteran pests to develop resistance to these chemicals. This situation underscores the urgent need for the development of new pest control methods that are both species-specific and environmentally friendly. Notably, the field of hemipteran biotechnology has made significant progress, with CRISPR/Cas-mediated hemipteran mutagenesis techniques gaining prominence [130].

#### 2.3.1. *Bemisia tabaci*

*B. tabaci* is a serious agricultural polyphagous pest and a carrier of several plant viruses that cause significant economic losses worldwide [131]. Traditional embryo injection methods are technically difficult and have high mortality rates, for example, the *A2083V* mutation associated with ketoenol resistance was detected using CRISPR/Cas9 introduced into *D. melanogaster* [112]. Thus, researchers have developed a new CRISPR/Cas9 gene-editing program based on the injection of vitellogenic adult females to form “ReMOT Control”. The protocol uses an ovary-targeting peptide ligand (BtKV) to fuse and inject Cas9 into adult females, enabling efficient, heritable editing of the offspring genome. This adult injection method is simple and easy to perform, requires no specialized equipment, and opens up new possibilities for gene-editing studies in *B. tabaci* [132].

#### 2.3.2. *Nilaparvata lugens*

*N. lugens* serves as the primary pest of rice crops in China and various Asian nations, relying exclusively on rice and wild rice for feeding and reproduction [133]. The resistance molecular mechanisms of *N. lugens* to multiple insecticides included P450-mediated metabolic detoxification as a major pathway. *CYP6ER1* overexpression is an important resistance mechanism of *N. lugens* to neonicotinoid insecticides. Zhang utilized CRISPR/Cas9 technology to create a *CYP6ER1*-knockout homozygous strain, revealing a significant increase in sensitivity to imidacloprid and thiacloprid, while the increase in sensitivity to other neonicotinoids was comparatively modest. Metabolic analyses and site predictions indicate that *CYP6ER1* activity correlates with insecticide structure, showing that imidacloprid and thiacloprid are readily oxidized in a specific five-membered heterocycle [110]. Zhang knocked out *CYP6CS1* to investigate its influence on resistance to various insecticides, finding a marked enhancement in sensitivity to imidacloprid and other insecticides. However, differences were noted in survival rates, reproduction, and weight between the CYP6CS1KO strain and the wild type [111]. A single non-synonymous SNP was found in the chitin synthase 1 gene (*CHS1*) at the amino acid position of 932, where a glycine (G) was replaced with cysteine (C), and scientists demonstrated that this single-amino acid replacement is associated with buprofezin resistance. At the same position, Zhang introduced the substitution mutations (*G932C*) of the CHS1 gene into *N. lugens* using the CRISPR/Cas9 system combined with the HDR pathway. The *N. lugens* lines that are homozygous for these mutations (*G932C*) were resistant to buprofezin [134].

#### 2.3.3. *Myzus persicae*

Aphidoidea is an insect that is a serious hazard to agriculture and requires effective control measures to reduce its damage. Simultaneously, Aphidoidea also has some special biological phenomena and ecological relationships that deserve further research and exploration [135]. Resistance of *M. persicae* to neonicotinoid insecticides is mainly caused by an arginine-to-threonine substitution at position 81 of the nAChR_beta 1 subunit (*R81T* mutation). This mutation confers tolerance to neonicotinoid insecticides in *M. persicae*. To validate the role of *R81T* in neonicotinoid resistance and to test whether it imposes any significant fitness costs on the insect, Homem used CRISPR/Cas9 to introduce a similar mutation in the genome of *D. melanogaster* and found that *D. melanogaster* carrying *R81T* showed increased resistance to neonicotinoid insecticides accompanied by a significant reduction in fitness [113]. In a follow-up study, a novel RPA-CRISPR/Cas12a-based rapid visual assay was developed to specifically target the nAChR subunit *V62I* and *R81T* mutations in *Aphis gossypii* to monitor its resistance to imidacloprid. The method uses *A. gossypii* tissue samples of less than 2 mm to accurately identify imidacloprid-resistant *A. gossypii* populations in the field in less than 1 h at 37 °C and 440–460 nm blue light [127].

## 3. Conclusions and Prospects

As the global population continues to grow and the need for food becomes more urgent, agricultural pests have become a serious challenge affecting food security and agricultural production. These pests not only cause heavy crop losses worldwide, but also pose a threat to human health by transmitting bacteria and viruses [136,137]. To address this challenge, scientists have been seeking effective pest control strategies. However, conventional insecticides have led to the gradual development of pest resistance due to overuse, which greatly reduces the effectiveness of pest management. Therefore, it has become particularly urgent to gain a deeper understanding of the mechanisms of insect resistance and to develop new pest control methods. The emergence of CRISPR/Cas technology has provided new perspectives and powerful tools for entomological research. Especially in exploring insect resistance, CRISPR/Cas technology not only helps to reveal the molecular mechanisms of insect resistance, but also offers the possibility of developing new pest management strategies.

In insect drug resistance research, CRISPR/Cas technology is widely used in a variety of insect models, such as *D. melanogaster*. Through CRISPR/Cas9 technology, researchers are able to precisely edit the genome of *D melanogaster* to explore the genes and pathways associated with drug resistance. For example, CYP genes, which are key members of insect metabolic detoxification enzymes, were identified in *D melanogaster* as important candidate genes associated with multiple insecticide metabolism and resistance [51,52]. In addition, nAChR and ABC transporter proteins also play important roles in insect resistance, and the knockdown or mutation of these genes by CRISPR/Cas9 technology can further reveal their resistance mechanisms [56,59]. In addition to *D melanogaster*, CRISPR/Cas technology has also been applied in the study of drug resistance in other agricultural pests, such as *H. armigera* [85,90], *S. exigua* [99,102], and *P. xylostella* [65,99]. In these pests, researchers knocked out resistance-related genes, such as cadherin, P450 gene, and GST gene, by CRISPR/Cas9 technology, thus revealing the role of these genes in pest resistance. In addition, the resistance mutants of pests can be created by CRISPR/Cas technology, providing new tools and methods for monitoring and management of pest resistance. By editing these gene, scientists can identify genes associated with insect resistance and, through the intensive study of these genes, further reduce the number of pests and the damage they cause to crops. In the control of public health pests, CRISPR technology also has broad application prospects. For example, *Ae. Aegypti* is a mosquito that is a major vector of serious infectious diseases such as dengue fever. With CRISPR technology, scientists can edit the genes of the *Ae. aegypti* to make it unable to carry and transmit viruses, thus blocking the spread of disease at the source [61,119].

CRISPR holds a substantial promise for leveraging its capabilities in insecticidal interventions and innovative approaches to pest management. In recent years, a team from the University of California has established a platform for precision-guided sterile insect technique (pgSIT) based on CRISPR and achieved remarkable results. They successfully created sterile insects by programming CRISPR/Cas9 to induce specific gene mutation sites, which resulted in a male infertility rate of over 99.5% and a female mortality rate of over 99.9% in hybrid offspring. This technique has been tested on a variety of pests including *D. melanogaster* and *An. gambiae*, among others [138,139]. DIPA-CRISPR (Direct Injection of Parental Animals for CRISPR) represents an innovative gene-editing technology. It involves directly injecting CRISPR/Cas9 components into the bodies of adult female insects. This method influences developing oocytes, allowing heritable genetic mutations in offspring, thereby simplifying the gene-editing process in insects [140]. However, to date, testing has occurred only in a limited number of insect species. Future advancements promise to enhance the efficiency and convenience of gene editing [141,142,143,144]. For populations that have developed resistance, CRISPR/Cas9 presents a potential solution—reversing resistance. Through gene drive technology, researchers can propagate specific genetic variations within insect populations, restoring sensitivity to insecticides [145]. Beyond reversing resistance, this technology also holds promise for the development of novel insecticides. By editing genes associated with drug metabolism, drug targets, or drug transport within insects, researchers can devise efficient, low-toxicity chemical or biological insecticides tailored to specific insect species [146]. Although the application of the CRISPR/Cas9 system to insecticide resistance in insects holds great promise, there are obvious challenges. One of the foremost concerns is off-target effects, whereby the system may inadvertently modify genes similar to the intended target. Researchers are adopting several strategies to address this issue, such as developing high-precision, low-off-target Cas9 variants and optimizing gRNA design with bioinformatics tools to ensure precise targeting. Technological developments have resulted in fewer off-target modifications, enhancing the safety and efficacy of applications [147].

Understanding the mechanisms by which insects develop resistance to insecticides can effectively prevent and control pests, significantly improving crop yield and quality, reducing crop losses, and ultimately increasing economic benefits. It can also reduce the number of pests such as mosquitoes and flies, thereby reducing the risk of disease transmission and ensuring human health [148,149].

## Figures and Tables

**Figure 1 insects-16-00345-f001:**
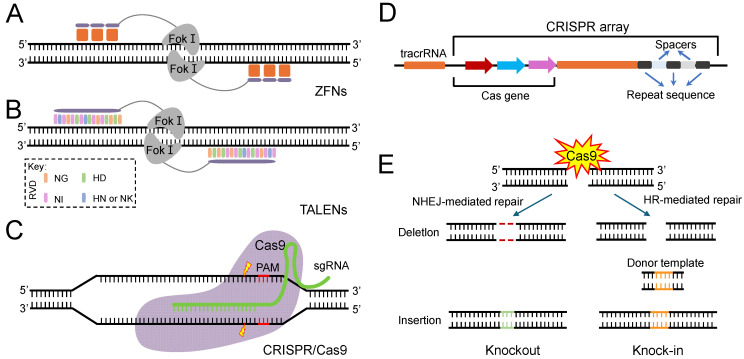
The architecture of ZFN, TALEN, and CRISPR/Cas systems. (**A**) The schematic of ZFN specifically recognizing and binding to DNA. Each DNA-recognition domain contains three zinc fingers, and each zinc finger structure is in direct contact with three bases. (**B**) The schematic of TALEN specifically recognizing and binding to DNA. The TALEN element is targeted and bound to specific DNA sites through a DNA-recognition module and then clipped at specific sites under the action of the Fok I nuclease. The RVD compositions are indicated in the dotted box (**C**) The schematic of cleavage by the Cas9 enzyme. The Cas9 enzyme recognizes the PAM (NGG) site and cleaves the target DNA sequence between the third and fourth bases near the PAM site. (**D**) The schematic of CRISPR site structure. The CRISPR/Cas system consists of CRISPR sequence elements and the Cas gene family. (**E**) The repair pathway of double-strand break (DSB) mediated by CRISPR system. The DSB induced by the Cas9/sgRNA complex can be repaired by non-homologous end joining (NHEJ) or homologous recombination (HR). This can result in small insertions or deletions at the target sites (left) and homologous repair with a desired template (right).

**Figure 2 insects-16-00345-f002:**
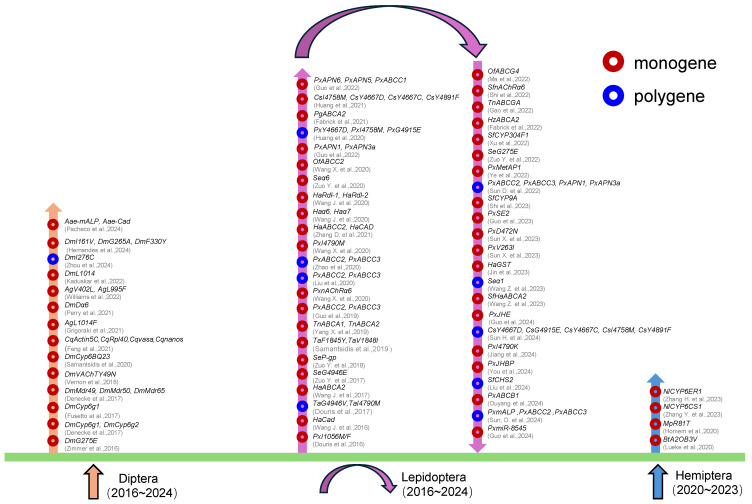
The application of CRISPR/Cas9 technology in insecticide resistance. Red circles represent single-gene knockout or knock-in, and blue circles represent multi-gene knockout. The orange arrow represents Diptera, the pink arrow represents Lepidoptera, and the light blue arrow represents Hemiptera. Diptera (Aa, *Aedes aegypti;* Ag, *Anopheles gambiae*; Cq, *Culex quinquefasciatus*; Dm, *Drosophila melanogaster*); Lepidoptera (Cs, *Chilo suppressalis;* Ha, *Helicoverpa armigera;* Hz, *Helicoverpa zea*; Of, *Ostrinia furnacalis*; Pg, *Pectinophora gossypiella*; Px, *Plutella xylostella;* Se, *Spodoptera exigua*; Sf, *Spodoptera frugiperda*; Ta, *Tuta absoluta*; Tn, *Trichoplusia ni*); and Hemiptera (Bt, *Bemisia tabaci*; Mp, *Myzus persicae*; Nl, *Nilaparvata lugens*). Sources Refs. [51,52,53,54,55,56,57,58,59,60,61,62,63,64,65,66,67,68,69,70,71,72,73,74,75,76,77,78,79,80,81,82,83,84,85,86,87,88,89,90,91,92,93,94,95,96,97,98,99,100,101,102,103,104,105,106,107,108,109,110,111,112,113].

**Table 1 insects-16-00345-t001:** Application of CRISPR/Cas9 system in insecticide resistance in insects.

Classification	Species	Genes	Mutation		Summary	References
**Diptera**	*Drosophila melanogaster*	*Cyp6g1*, *Cyp6g2*	Knockout	monogenic	Only high expression of *Cyp6g1* contributes to imidacloprid resistance, while overexpression of *Cyp6g2* can metabolize imidacloprid and produce resistance.	Denecke + 2017 [51]
		*Cyp6g1*	Knockout	monogenic	The *Cyp6g1* gene is associated with the oxidative metabolism of insecticides but does not directly control the production of nitro-reducing metabolites.	Fusetto + 2017 [52]
		*L1014*	Knock-in	monogenic	The CRISPR-based allele drive replaces the resistant *kdr* mutation with a susceptible wild-type counterpart.	Kaduskar + 2022 [53]
		*Cyp6BQ23*	Knockout	monogenic	Reduced pyrethroid affinity at the target site, delaying saturation while simultaneously extending the duration of P450-driven detoxification	Samantsidis + 2020 [54]
		*G275E*	Knock-in	monogenic	The *G275E* gene was associated with spinosad resistance using the CRISPR/Cas9 system.	Zimmer + 2016 [55]
		*Dα6*	Knockout	monogenic	The mechanism of nAChR’s response to three different chemical classes of insecticides was explored by CRISPR.	Perry + 2021 [56]
		*I161V*, *G265A*, *F330Y*	Knockout	monogenic	Insecticide susceptibility gene drives could be useful tools to control pest insects; however, problems with particularities of target loci and GDR will need to be overcome for them to be effective.	Hernandes + 2024 [57]
		*VAChT Y49N*	Knock-in	monogenic	The GAL4/UAS system was combined with CRISPR technology to study the effect of *VAChTY49N* on CASPP resistance.	Vernon + 2018 [58]
		*Mdr49*, *Mdr50*, *Mdr65*	Knockout	monogenic	Demonstrated the role of ABC transporter in insecticide tolerance of *D. melanogaster*	Denecke + 2017 [59]
		*I276C*	Knockout	polygenic	The intersubunit amino acids within the transmembrane M1 and M3 domains do form binding sites that are critical for the interaction of m-diamide and isoxazoline insecticides.	Zhou + 2024 [60]
	*Aedes aegypti*	*Aae-mALP*, *Aae-Cad*	Knockout	monogenic	In addition to *Aae-Cad* and *Aae-mALP*, other midgut membrane proteins are also involved in the Cry toxin receptor mode of Aspergillus aegypti.	Pacheco + 2024 [61]
	*Culex quinquefasciatus*	*Actin5C*, *Rpl40*, *vasa*, *nanos*	Knockout	monogenic	The gRNA scaffold variant improved the transgene efficiency of Culex mosquitoes.	Feng + 2021 [62]
	*Anopheles gambiae*	*L1014F*	Knock-in	monogenic	Mosquitoes carrying the *L1014F* allele show an adaptive disadvantage in the homozygous state.	Grigoraki + 2021 [63]
		*V402L*, *L995F*	Knock-in	monogenic	In some cases, the lower fitness costs associated with this kdr mutation may have a selective advantage over classical kdr.	Williams + 2022 [64]
**Lepidoptera**	*Plutella xylostella*	*ABCC2*, *ABCC3*	Knockout	monogenic	Two knockout strains were successfully constructed using the new CRISPR/Cas9 genome engineering system.	Guo+ 2019 [65]
		*APN6*, *APN5*, *ABCC1*	Knockout	polygenic	Reveals how *P. xylostella* adjusts MAPK phosphorylation in response to toxins and alters FTZ-F1 transcription factor binding to regulate the expression of Bt receptors or non-receptor paralogues	Guo + 2022 [66]
**Lepidoptera**	*Plutella xylostella*	*APN1*, *APN3a*	Knockout	monogenic	Evidence that the MAPK cascade response can be activated by enhanced upstream hormone signaling to counter Bt virulence in the diamondback moth.	Guo + 2022 [67]
		*ABCC2*, *ABCC3* *APN1*, *APN3a*	Knockout	polygenic	Bt toxins have multiple modes of action that can compensate for the loss of a single receptor.	Sun D. + 2022 [68]
		*PxmALP*, *PxABCC2*, *PxABCC3*	Knockout	polygenic	Reveals functional redundancy between ABC transporter proteins and *PxmALP*	Sun D. + 2024 [69]
		*PxABCC2*, *PxABCC3*	Knockout	polygenic	*PxABCC2* and *PxABCC3* are redundant or complementary.	Liu + 2020 [70]
		*PxABCC2*, *PxABCC3*	Knockout	polygenic	The value of using single-gene knockout and multi-gene knockout is emphasized.	Zhao + 2020 [71]
		*PxABCB1*	Knockout	monogenic	*PxABCB1* protects insects from avermectin insecticides; on the other hand, it promotes the toxic effects of Bt Cry1Ac toxin.	Ouyang + 2024 [72]
		*PxMetAP1*	Knockout	monogenic	Revealed the important role of the MetAP gene in DBMBt tolerance	Ye + 2022 [73]
		*PxJHBP*	Knockout	monogenic	*PxJHBP* is a key gene in resistance to Cry1Ac and regulation of female reproduction.	You + 2024 [74]
		*SE2*	Knockout	monogenic	Adoption of high-efficiency double sgRNA strategy	Guo + 2023 [75]
		*miR-8545*	Knockout	monogenic	Increased expression of microRNA (*miR-8545*) inhibits the newly discovered molting steroid degrading enzyme (PxGLD)	Guo + 2024 [76]
		*I1056M/F*	Knockout	monogenic	With the *D. Melanogaster* model, scientists were able to quickly observe phenotypic changes after gene editing, thus verifying the effect of the mutation.	Douris et al. + 2016 [77]
		*PxJHE*	Knockout	monogenic	CRISPR/Cas9-induced m^6^A site-specific mutation *PxJHE* induces fitness costs	Guo + 2024 [78]
		*nAChRα6*	Knockout	monogenic	Endogenous functional studies demonstrated the causal relationship of *Pxα6* truncated mutations.	Wang + 2020 [79]
		*V263I*	Knockout	monogenic	The function of *V263I* mutation in PxGluCl was verified for the first time	Sun + 2023 [80]
		*D472N*	Knockout	monogenic	The homozygous *D472N* mutation in Rdl1 confers a low level of resistance to avermectin in *P. xylostella*	Sun X. + 2023 [81]
		*I4790M*	Knock-in	monogenic	The functional role of PxRyR’s *I4790M* mutation in diamide resistance was confirmed for the first time.	Wang X + 2020 [82]
**Lepidoptera**	*Plutella xylostella*	*I4790K*	Knock-in	monogenic	The *I4790K* mutation reduces insecticide binding to receptors	Jiang + 2024 [83]
		*Y4667D*, *I4758M*, *G4915E*	Knock-in	monogenic	Multiple mutations in *RyR* confer resistance to diamides in clostridium-inhibiting bacteria.	Huang + 2020 [84]
	*Helicoverpa armigera*	*HaCad*	Knockout	monogenic	*HaCad* provides strong reverse genetic evidence as a functional receptor of Cry1Ac.	Wang + 2016 [85]
		*HaRdl-1*, *HaRdl-2*	Knockout	monogenic	*HaRdl-1* and *HaRdl-2* are important determinants of *H. armigera* sensitivity to three cyclodiene insecticides.	Wang J. + 2020 [86]
		*Haα6*, *Haα7*	Knockout	monogenic	Variation in *nAChRα6* was associated with high resistance of pests to spinosyns.	Wang J. + 2020 [87]
		*HaABCA2*	Knockout	monogenic	The midgut brush marginal membrane vesicles of knockout populations lost their ability to bind to Cry2Ab, but retained their ability to bind to Cry1Ac.	Wang J. + 2017 [88]
		*HaABCC2*, *HaCAD*	Knockout	monogenic	The synergistic effect of CAD and *ABCC2/ABCC3* significantly enhanced the resistance of *H. armigera* to Cry1Ac.	Zhang D. + 2021 [89]
		*GST*	Knockout	polygenic	The complex changes in GST cluster expression enhanced resistance of field populations to the highly efficient pyrethroid.	Jin + 2023 [90]
	*Pectinophora gossypiella*	*PgABCA2*	Knockout	monogenic	Demonstrated that destructive mutations lead to actual resistance to Cry2Ab and are associated with field resistance	Fabrick + 2021 [91]
	*Helicoverpa zea*	*HzABCA2*	Knockout	monogenic	The mutation of the *HzABCA2* gene is a key factor leading to the development of resistance to Cry2Ab in *Helicoverpa zea*.	Fabrick + 2022 [92]
	*Spodoptera frugiperda*	*CYP304F1*	Knockout	monogenic	*CYP304F1* plays an important role in resistance to β-cypermethrin and chlorpyrifos.	Xu + 2022 [93]
		*SfCHS2*	Knockout	polygenic	Identified for the first time that the LTR retrotransposon Yaoer plays a pivotal role in the resistance mechanism against Vip3Aa	Liu + 2024 [94]
		*nAChRα6*	Knockout	polygenic	The team used CRISPR to knockout Sfα6 in *S. frugiperda*, studying its role in spinosyn susceptibility.	Shi + 2022 [95]
		*CYP9A*	Knockout	polygenic	*CYP9A* gene cluster knockout in *S.exigua* and *S. frugiperda*	Shi + 2023 [96]
	*Trichoplusia ni*	*ABCA1*, *ABCA2*	Knockout	monogenic	*ABCA2* is critical to the toxicity of Cry2Ab in T. ni.	Yang X. + 2019 [97]
		*ABCC2*	Knock-in	monogenic	Investigated the association between *ABCC2* and Trichoplusia ni resistance	Ma X. + 2022 [98]
	*Spodoptera exigua*	*G4946E*	Knock-in	monogenic	The *G4946E* mutation in the RyR gene was shown to confer a high level of resistance to diamide insecticides.	Zuo Y. + 2017 [99]
		*SeP-gp*	Knockout	monogenic	Overexpression of *SeP-gp* may lead to abamectin and EB resistance in S.exigua.	Zuo Y. + 2018 [100]
		*Seα6*	Knockout	monogenic	Proved the functional role of Seα6 in the treatment of spinosad and spinetoram	Zuo Y + 2020 [101]
		*G275E*	Knock-in	monogenic	Verified the role of *G275E* mutation in *S. exiguan* AChRα6 in resistance to spinosyn	Zuo Y. + 2022 [102]
		*Seα1*	Knockout	monogenic	*Seα1* knockout results in the loss of functional transmembrane (TM)3 and TM4 elements.	Wang Z. + 2023 [103]
	*Ostrinia furnacalis*	*ABCC2*	Knockout	monogenic	The OfABCC2 protein may function as a receptor for Cry1Fa, enhancing its association with Cry1Fa toxin’s mode of action.	Wang X. + 2020 [104]
		*ABCG4*	Knockout	monogenic	The mutant exhibited enhanced Cry1 toxin resistance, impacting larval development and population.	Gao + 2022 [105]
**Lepidoptera**	*Chilo suppressalis*	*I4758M*, *Y4667D*, *Y4667C*, *Y4891F*	Knock-in	monogenic	Revealed the role of RyR mutation in diamide resistance and how the mutation affects the binding affinity of different diamides	Huang + 2021 [106]
		*Y4667D*, *G4915E*, *Y4667C*, *I4758M*, *Y4891F*	Knockout	polygenic	The *I4758M* and *Y4667C* double mutations have higher tetraniliprole resistance than the single *Y4667C* mutation.	Sun H. + 2024 [107]
	*Tuta absoluta*	*F1845Y*, *V1848I*	Knockout	monogenic	The *V1848I* and *F1845Y* mutations may have formed too large a side chain to affect metaflumizone binding.	Samantsidis + 2019 [108]
		*G4946V*, *I4790M*	Knock-in	monogenic	Confirmed the role of RyR mutations in diamide resistance and revealed how mutations affect the binding affinity of different diamides	Douris et al. + 2017 [109]
**Hemiptera**	*Nilaparvata lugens*	*CYP6ER1*	Knockout	monogenic	*CYP6ER1* activity is related to the structure of an insecticide.	Zhang H. + 2023 [110]
		*NlCYP6CS1*	Knockout	monogenic	Nl6CS1KO is similar to the wild type in development and longevity, but there are differences in survival, reproduction, and body weight.	Zhang Y. + 2023 [111]
	*Bemisia tabaci*	*A2083V*	Knockout	monogenic	*B. tabaci* is highly resistant to ketoenol insecticides, which is not related to metabolic resistance but is caused by *A2083V* mutation in the CT domain of ACC.	Lueke + 2020 [112]
**Hemiptera**	*Myzus persicae*	*R81T*	Knockout	monogenic	Introduction of the *R81T* mutation in the black-bellied fly maggot using CRISPR/Cas9 results in enhanced resistance to neonicotinoid insecticides but reduced fitness.	Homem + 2020 [113]

## Data Availability

Data are contained within the article.

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
