# Peer review of "CRISPR/Cas Technology in Insect Insecticide Resistance"

_insects, 2025, doi:10.3390/insects16040345_

Round 1

Reviewer 1 Report

Comments and Suggestions for Authors

Commnets

Title:The current title could not correctly reflect the content of the manuscript. There are severeal reverse genetics strategies for understanding the insecticide resistance in insects.

Abstract: It points that the insecticides spraying is still the crucial ways for gurarnteeing the agricultural and forestry production, therefore, the  main body should be focused more on the insecticide resistance of agricultural and forsery insects.

Introduction: Too much descripiation on the principals of the gene editing technologies, such as CRISPR/Cas system. There were lot of references illustrated this content, in my opinion it is unnecessariate. It's better to illustrate the advantages of genome editing technologies on understanding the insecticide resistance in insects than other reverse genetics strategies.

Applications of CRISPR/Cas9 in insecticide resistance: 1. focused more on the insecticide resistance of agricultural and forsery insects. Suggestions: preparing two seperate parts: genome editing technologies in agricultural pests and public healthy pests, respectively.

                                                                                        2. Only illustrated the previous experiemental results.

                                                                                        3. Lacking the summarize of previous results.

                                                                                        4. Too many XX. et al. at the start of sentences.

                                                                                        5. Personally suggestion: deleted the content of B. mori.

Discussion: This part is unnecessariate. You could arrange that contents into the previous sections. 

Other suggestion:The Supplementary file is the same table of as the main text. Please delete it or prepare some other content for Supplementary file.

Author Response

Dear Reviewer,

We would like to thank you for the time spent on our behalf reviewing the manuscript insects-3369143 entitled “Application of CRISPR/Cas Technology in Insect Insecticide Resistance”. The insightful and constructive comments from expert reviewer has been very helpful to address some issues with the previous version and improve the manuscript. We have corrected the errors and carefully revised the manuscript according to the reviewer’s suggestions, and we consider that our explanations and actions to the questions raised are reasonable. Moreover, we have incorporated most of the suggested changes in the revised manuscript and have highlighted them in yellow. The following is a point-bypoint response to the reviewers’ comments:

Reviewer 1

Comment 1: Title:The current title could not correctly reflect the content of the manuscript. There are severeal reverse genetics strategies for understanding the insecticide resistance in insects.

Response 1: Thank you for your comments and providing good suggestions for my manuscript. The title has been changed to "Application of CRISPR/Cas Technology in Insect Insecticide Resistance"(Lines 2-3).

Comment 2: Abstract: It points that the insecticides spraying is still the crucial ways for gurarnteeing the agricultural and forestry production, therefore, the main body should be focused more on the insecticide resistance of agricultural and forsery insects.

Response 2: Thank you very much for your valuable comments. We have revised the abstract section to focus more on insecticide resistance (Lines 18-24).

Comment 3: Introduction: Too much descripiation on the principals of the gene editing technologies,such as CRISPR/Cas system. There were lot of references illustrated this content, in my opinion it is unnecessariate. It's better to illustrate the advantages of genome editing technologies on understanding the insecticide resistance in insects than other reverse genetics strategies.

Response 3: Thank you for the comments. The relevant paragraphs have been deleted accordingly, and we added advantages of genome editing technology in understanding insecticide resistance (Lines 72-102).

Comment 4: Focused more on the insecticide resistance of agricultural and forsery insects. Suggestions: preparing two seperate parts: genome editing technologies in agricultural pests and public healthy pests, respectively.

Response 4: Thank you for your suggestion. Since the main body of the article is narrated according to different insect orders, and public health pests do not account for a large proportion, the narrative structure would be somewhat unbalanced if divided into these two parts. In response to your constructive suggestions, we have decided to discuss the article in two separate sections during our discussion (Lines 511-524).

Comment 5: Only illustrated the previous experiemental results. Lacking the summarize of previous results.

Response 5: Thank you for your suggestion. Relevant changes have been made in the article (Lines 199-203, 237-240, 290-293, 375-382).

Comment 6: Too many XX. et al. at the start of sentences.

Response 6: Most of the sentences on et al have been changed (Lines 68-69, 155-156, 314-317 and so on).

Comment 7: Personally suggestion: deleted the content of B. mori.

Response 7: Thank you for your suggestion. Revised accordingly.

Comment 8: Discussion: This part is unnecessariate. You could arrange that contents into previous sections.

Response 8: The relevant discussion section has been removed and added to the Introduction section (Lines 117-130).

Reviewer 2 Report

Comments and Suggestions for Authors

1.      The Table 1, line 140, is a bit confusing because some columns are too close together. I suggest formatting the columns to align them to the left.

2.      The section 2.2 Lepidoptera, lines 314-315, I suggest separating the species a little more. Just as was done in sections 2.1. Diptera, and 2.3 Hemiptera. I understand that there are many species of Lepidoptera that are being discussed, but the paragraphs are too long and talk about many different species. I suggest grouping them into subsections grouped by genus or biological semblances. Example:

2.2 Lepidotera

               B. mori

               H. armigera and H. zea (they are both Helicoverpa)

               Pectinophora gossypiella

               Trichoplusia ni and P. xylostella (they are both pinworms )

               S. frugiperda, S. exigua and Spodoptera litura (They are all Spodoptera)

               Ostrinia furnacalis

3.      The reverse genetics strategy is a highly relevant method, and therefore, Section 4, 'Perspectives and Future Directions,' could benefit from focusing more on the future prospects for insect pest management. After all, you are publishing in the 'Insect Pest and Vector Management' section of the journal. You have discussed several insect pests in your manuscript, so it would be valuable to consolidate this information and include a brief paragraph on potential strategies for controlling these pests. You could also highlight the benefits that effective pest control could bring to agriculture, such as improved crop yields and reduced losses.

 [

Author Response

Dear Reviewer,

We would like to thank you for the time spent on our behalf reviewing the manuscript insects-3369143 entitled “Application of CRISPR/Cas Technology in Insect Insecticide Resistance”. The insightful and constructive comments from expert reviewer has been very helpful to address some issues with the previous version and improve the manuscript. We have corrected the errors and carefully revised the manuscript according to the reviewer’s suggestions, and we consider that our explanations and actions to the questions raised are reasonable. Moreover, we have incorporated most of the suggested changes in the revised manuscript and have highlighted them in yellow. The following is a point-bypoint response to the reviewers’ comments:

Comment 1: The Table 1, line 140, is a bit confusing because some columns are too close together. I suggest formatting the columns to align them to the left.

Response 1: Thanks very much for taking your time to review this manuscript. Changes have been made to the form as requested.

Comment 2: The section 2.2 Lepidoptera, lines 314-315, I suggest separating the species a little more. Just as was done in sections 2.1. Diptera, and 2.3 Hemiptera. I understand that there are many species of Lepidoptera that are being discussed, but the paragraphs are too long and talk about many different species. I suggest grouping them into subsections grouped by genus or biological semblances.

Response 2: Thanks to your suggestion, sections 2.3 has been reclassified, with only one entry for Pectinophora gossypiella and therefore merged with H. armigera, Helicoverpa zea (Lines 253-445).

Comment 3: The reverse genetics strategy is a highly relevant method, and therefore, Section 4, 'Perspectives and Future Directions,' could benefit from focusing more on the future prospects for insect pest management. After all, you are publishing in the 'Insect Pest and Vector Management' section of the journal. You have discussed several insect pests in your manuscript, so it would be valuable to consolidate this information and include a brief paragraph on potential strategies for controlling these pests. You could also highlight the benefits that effective pest control could bring to agriculture, such as improved crop yields and reduced losses.

Response 3: Thank you very much for your helpful comments regarding this manuscript. The content has been added to the discussion section (Lines 506-563).

Reviewer 3 Report

Comments and Suggestions for Authors

The authors have done a good literature review and have identified many studies where genome editing has been applied either to study the role of genes and mutations in insecticide resistance or to gain better insight into the insecticides’ mode of action. The figures (upon editing – comments provided below) and Table are informative and a good way to summarize the current knowledge. However, the main text needs substantial editing. In many cases the text feels more like a listing of genes that have been genome edited. Thus, in many cases the text lacks coherence and flow. There are also some cases where the information of the provided reference studies has been misunderstood and some cases where the information provided is irrelevant to insecticides and resistance. Before the manuscript can be considered for publication a careful editing needs to be done to address those points. It would also help to split the paragraphs further or rearrange them to provide context to each paragraph. For example, LINES 146-221 include information about many different detox genes and target site mutations that are not presented in a specific order. Below are some more comments.   

Introduction: the two different insecticide resistance mechanisms A. target site resistance (related to the insecticide’s target protein e.g mutations) and B. metabolic resistance involving detoxification enzymes are not clearly separated and presented, but rather confused.

Figure 1, The legend needs editing. The boxes with different colors in the ZNF do not represent  DNA bases, as they are depicted as part of the protein. The colored boxes in the TALEN DNA binding domains are not explained. The Cas9 does not recognize only the PAM but rather  a 20bp sequence followed by a PAM , to which it is guided by the guide RNA.  

Figure 2. The figure has some nice features, but can also be confusing. The green arrowhead gives the impression there is a chronological order in the studies presented, which is not true. Also, it is not clear why the arrows for diptera and lepidoptera are split in two. Why is one of the arrows facing up and the other down? Is there a meaning for that? I think it would help if the species name is also provided in front of each gene name that was edited

Some examples where language editing is required

Line 13”  is the expression alterations”

Line 30:” as vectors or carriers to transmit a variety of bacteria and viruses”

Line 66-68:” Although  ZFNs played a key role in early insect gene editing research, however, relatively complex  preparation processes and high costs have largely limited their widespread application

Line 14-15: the definition of target genes given is not correct. The sentence needs to be rephrased.

Line 75:” the unit assembly strategy is not explained properly and it is difficult for the  reader to follow the information provided.

Lines 73-84: the text here needs substantial editing so that the information is well presented and easy for the reader to follow.

Line 44-45:” and CYP6F1 serves as a crucial element in the insect resistance mechanism of Culex pipiens pallens by enhancing  expression levels of insecticides such as deltamethrin” this doesn’t make sense

Line 52-54. Here again the sentence is wrong as target proteins do not detoxify insecticide molecules

Lines 56-58: what does environmentally friendly mean?

Lines 123-124. What do you mean by integrating the relationship between insect resistance and CRISPR/Cas9 technology? There is no relationship.

Lines 156-158:”  is associated with the formation of insecticide oxidative metabolites, whereas the production of nitro-reduced metabolites may be controlled by microbial activity, rather than directly regulated by CYP6G1. The meaning is not clear. There seems to be something missing from the info provided.

Line 180. The conclusion of the study (ref 57) is missing (this applies in other cases as well, please check)

Line 198:” F1845Y and V1848I are two mutations in the sodium channel domain IVS6 segment” In which species?

Line 210:;” Mutations of G4946E/V and I4790M in RyRs confer resistance to diamide insecticides” in which species?

Lines 232-237: this is irrelevant to the topic of insecticide resistance

Lines 254-257: the studies (ref 76 and 77) were not performed on An.stefensi but An. Gambiae

Lines 261-274: the information provided as such is irrelevant. They could be included but under a different title.

Lines 276-290: irrelevant to insecticide resistance

Lines 357-365: the studies presented  provide insight into the mode of action of the toxins and insecticides, but it is not clearly explained how they will  facilitate monitoring and management of resistance to those toxins/insecticides 

Lines 375-378 and 404-406. The information is not properly connected

Lines 407-426: this is an example of a paragraph where the sentences are not connected. There is just a listing of different information.

Lines 483-484: the statement is not correct

Lines 555-556:” Beyond reversing resistance, this technology also holds promise for the development of novel insecticides”. It is not clearly explained how genome editing will facilitate the development of novel insecticides.

Lines 569-572: This is not correctly written the SIT is causing only male sterility. The pgSIT led to over 99.5% male sterility and over 99.9% female mortality in the hybrid offspring, as shown in the ref provided below

Comments on the Quality of English Language

Language editing is required. I have provided a few only examples of text that needs editing in the section "comments and suggestions for Authors"

Author Response

Dear Reviewer,

We would like to thank you for the time spent on our behalf reviewing the manuscript insects-3369143 entitled “Application of CRISPR/Cas Technology in Insect Insecticide Resistance”. The insightful and constructive comments from expert reviewer has been very helpful to address some issues with the previous version and improve the manuscript. We have corrected the errors and carefully revised the manuscript according to the reviewer’s suggestions, and we consider that our explanations and actions to the questions raised are reasonable. Moreover, we have incorporated most of the suggested changes in the revised manuscript and have highlighted them in yellow. The following is a point-bypoint response to the reviewers’ comments:

Comment 1: The authors have done a good literature review and have identified many studies where genome editing has been applied either to study the role of genes and mutations in insecticide resistance or to gain better insight into the insecticides’ mode of action. The figures (upon editing – comments provided below) and Table are informative and a good way to summarize the current knowledge. However, the main text needs substantial editing. In many cases the text feels more like a listing of genes that have been genome edited. Thus, in many cases the text lacks coherence and flow. There are also some cases where the information of the provided reference studies has been misunderstood and some cases where the information provided is irrelevant to insecticides and resistance. Before the manuscript can be considered for publication a careful editing needs to be done to address those points. It would also help to split the paragraphs further or rearrange them to provide context to each paragraph. For example, LINES 146-221 include information about many different detox genes and target site mutations that are not presented in a specific order. Below are some more comments.

Response 1: Thank you very much for your kindness comments and for giving us the opportunity to revise the manuscript The information unrelated to insecticide resistance has been deleted and the erroneous information has been corrected. We have incorporated 2.2 Split Lepidoptera chapters for better reading (Lines 146-201 253-445).

Comment 2: Figure 1, The legend needs editing. The boxes with different colors in the ZNF do not represent DNA bases, as they are depicted as part of the protein. The colored boxes in the TALEN DNA binding domains are not explained. The Cas9 does not recognize only the PAM but rather a 20bp sequence followed by a PAM , to which it is guided by the guide RNA.

Response 2: Thank you very much for your valuable comments. The figure 1 has been revised accordingly (Lines 104-109).

Comment 3: Figure 2. The figure has some nice features, but can also be confusing. The green arrowhead gives the impression there is a chronological order in the studies presented, which is not true. Also, it is not clear why the arrows for diptera and lepidoptera are split in two. Why is one of the arrows facing up and the other down? Is there a meaning for that? I think it would help if the species name is also provided in front of each gene name that was edited

Response 3: Thank you very much for your valuable comments. We have revised Figure 2 and figure legends to make it clearer (Line 139-143).

Comment 4: Some examples where language editing is required. Line 13”is the expression ”

Response 4:Revised accordingly (Lines 21-24).

Comment 5: Line 30:” as vectors or carriers to transmit a variety of bacteria and viruses”

Response 5:Revised accordingly (Lines 42-44).

Comment 6: Line 66-68:” Although ZFNs played a key role in early insect gene editing research, however, relatively complex preparation processes and high costs have largely limited their widespread application

Response 6:Revised accordingly (Lines 72-74).

Comment 7: Line 14-15: the definition of target genes given is not correct. The sentence needs to be rephrased.

Response 7:Revised accordingly (Lines 21-24).

Comment 8: Line 75:” the unit assembly strategy is not explained properly and it is difficult for the reader to follow the information provided.

Comment 8: Thank you very much for your valuable comments. Explanation has been added (Lines 79-82).

Comment 9: Lines 73-84: the text here needs substantial editing so that the information is well presented and easy for the reader to follow.

Comment 9: Revised accordingly (Lines 72-84).

Comment 10: Line 44-45:” and CYP6F1 serves as a crucial element in the insect resistance mechanism of Culex pipiens pallens by enhancing expression levels of insecticides such as deltamethrin” this doesn’t make sense

Comment 10: Revised accordingly (Lines 57-58).

Comment 11: Line 52-54. Here again the sentence is wrong as target proteins do not detoxify insecticide molecules

Comment 11: Revised accordingly

Comment 12: Lines 56-58: what does environmentally friendly mean?

Comment 12: Deleted accordingly.

Comment 13: Lines 123-124. What do you mean by integrating the relationship between insect resistance and CRISPR/Cas9 technology? There is no relationship.

Comment 13: Revised accordingly (Lines 122-124).

Comment 14: Lines 156-158:”is associated with the formation of insecticide oxidative metabolites, whereas the production of nitro-reduced metabolites may be controlled by microbial activity, rather than directly regulated by CYP6G1. The meaning is not clear. There seems to be something missing from the info provided.

Comment 14: Revised accordingly (Lines 150-156).

Comment 15: Line 180. The conclusion of the study (ref 57) is missing (this applies in other cases as well, please check)

Comment 15: Revised accordingly (Lines 180-188).

Comment 16: Line 198:” F1845Y and V1848I are two mutations in the sodium channel domain IVS6 segment” In which species?

Comment 16: Thank you for your careful review and constructive suggestions regarding our manuscript. F1845Y and V1848I are two mutations in Plutella xylostella and Tuta absoluta. Revised accordingly (Lines 356-361).

Comment 17: Line 210: “Mutations of G4946E/V and I4790M in RyRs confer resistance to diamide insecticides” in which species?

Comment 17: Thank you for your careful review and constructive suggestions regarding our manuscript. G4946V and I4790M are two mutations in Tuta absoluta. Revised accordingly (Lines 366-368).

Comment 18: Lines 232-237: this is irrelevant to the topic of insecticide resistance

Comment 18: Deleted accordingly.

Comment 19: Lines 254-257: the studies (ref 76 and 77) were not performed on An.stefensi but An. Gambiae

Comment 19: Revised accordingly (Lines 230-235).

Comment 20: Lines 261-274: the information provided as such is irrelevant. They could be included but under a different title.

Comment 20: Deleted accordingly.

Comment 21: Lines 276-290: irrelevant to insecticide resistance

Comment 21: Deleted accordingly.

Comment 22: Lines 357-365: the studies presented provide insight into the mode of action of the toxins and insecticides, but it is not clearly explained how they will facilitate monitoring and management of resistance to those toxins insecticides.

Comment 22: Revised accordingly (Lines 403-407).

Comment 23: Lines 375-378 and 404-406. The information is not properly connected

Comment 23: Revised accordingly (Lines 336-338).

Comment 24: Lines 407-426: this is an example of a paragraph where the sentences are not connected. There is just a listing of different information.

Comment 24: Revised accordingly (Lines 341-374).

Comment 25: Lines 483-484: the statement is not correct

Comment 25: Deleted accordingly.

Comment 26: Lines 555-556:” Beyond reversing resistance, this technology also holds promise for the development of novel insecticides”. It is not clearly explained how genome editing will facilitate the development of novel insecticides.

Comment 26: The content has been added to the discussion section (Lines 547-551).

Comment 27: Lines 569-572: This is not correctly written the SIT is causing only male sterility. The pgSIT led to over 99.5% male sterility and over 99.9% female mortality in the hybrid offspring, as shown in the ref provided below

Comment 27: Revised accordingly (Line 527-531).

Round 2

Reviewer 1 Report

Comments and Suggestions for Authors

1. Please check the format of all latin name in the entire text.

2. The mosquitoes section need a clearly and good summary. Please restructure the existed one.

3. Discussion title is seldom present  in the review article. The last section of most review articles are conclusion and prospects. Please restructure the contents according to the title of conclusion and prospects.

Comments on the Quality of English Language

The English should be improved by native experts.

Author Response

Editor

Insects

January 18, 2025

Dear Reviewer,

We would like to thank you reviewers for the time spent on our behalf reviewing the manuscript insects-3369143 entitled “Application of CRISPR/Cas Technology in Insect Insecticide Resistance”. The insightful and constructive comments from expert reviewers have been very helpful to address some issues with the previous version and improve the manuscript. We have corrected the errors and carefully revised the manuscript according to the reviewer’s suggestions, and we consider that our explanations and actions to the questions raised are reasonable. Moreover, we have incorporated most of the suggested changes in the revised manuscript and have highlighted them in yellow. The following is a point-bypoint response to the reviewers’ comments:

Comments from the editors and reviewers:

Reviewer 1

Comment 1: Please check the format of all latin name in the entire text.

Response 1: Thank you very much for your suggestion. We have checked the entire Latin name and found several errors (Lines 148 250 251 426 458 495).

Comment 2: The mosquitoes section need a clearly and good summary. Please restructure the existed one.

Response 2: Thank you for your comments and providing good suggestions for my manuscript. The Mosquitoes section has been revised as required (Lines 206-238).

Comment 3: Discussion title is seldom present in the review article. The last section of most review articles are conclusion and prospects. Please restructure the contents according to the title of conclusion and prospects.

Response 3: Thank you very much for your valuable comments. The conclusion and prospects section has been added as required (Lines 506-573).